# Optical Frequency Comb-Based Direct Two-Photon Cooling for Cold Atom Clock

Lin Dan [ID], Hao Xu , Ping Guo and Jianye Zhao *

Department of Electronics, Peking University, Beijing 100871, China; l_d@pku.edu.cn (L.D.);
hall_xu@pku.edu.cn (H.X.); pingguo@pku.edu.cn (P.G.)
* Correspondence: zhaojianye@pku.edu.cn; Tel.: +86-010-62754253

**Abstract:** The performance of the cold atom clock based on coherent population trapping (CPT) improved when the temperature decreased. In order to obtain a lower temperature in the cold atom clock, we proposed a cooling scheme in this paper that employs direct two-photon transition using optical frequency combs (OFCs). Two trains of time-delayed pulses from opposite directions were utilized to interact with atoms. It was found that the temperature of the cold atoms reached the minimum if the pulse area was π and the time delay between the absorption pulse and the stimulated emission pulse was in the range from $0.7\tau$ to $\tau$. In this paper, it was confirmed that the proposed cooling process allowed for faster and more efficient momentum exchange between light and atoms, and the proposed cooling process could be applied to the atoms or molecules that could not be cooled to desired temperature through the single-photon cooling process. The $^{87}$Rb cooling, together with the CPT interrogating scheme using OFCs reduced the ratio value of linewidth/contrast, and the frequency stability of the cold atom clock hence improved by more than six times as per our calculation.

**Keywords:** CPT; OFCs; direct two-photon transition; stimulated emission; cold atom clock





## 1. Introduction

The performance of the atomic frequency standard based on the coherent population trapping (CPT) using vapour-cell alkali metal atoms has greatly improved in stability and accuracy [1–3]. Several optimization schemes based on proposed CPT were aimed at improving the contrast of the signal, thereby improving the frequency stability, such as push-pull [4], σ⁺-σ⁻ [5], lin//lin [6], and lin⊥lin [7]. Field applications of chip-scale atomic clocks (CSACs) [8,9] based on CPT are growing rapidly because of their small size and low power consumption. Typically, CSACs deliver short-term fractional frequency stability of $<1 \times 10^{-10}$ [10], but the devices are substantively not accurate because of large systematic frequency shifts derived from high-pressure buffer gases and light shifts [11]. The temperature coefficient for the change in fractional frequency of high-performance buffer-gas CPT clocks has been measured to be in the order of $10^{-10}$/K [2,12,13].

The frequency shifts and associated instability derived from buffer gases can be eliminated by performing CPT with cold atoms [14–16]. Using laser-cooled atoms can eliminate Doppler broadening, thus narrowing the atoms' optical spectra and creating a clean system in which all cold atoms are uniformly interrogated and light shifts can be precisely studied [15].

A typical Λ system for CPT interrogation is shown in Figure 1. The atoms are evenly distributed in the ground states $|1\rangle$ and $|2\rangle$, and the bichromatic laser (with angular frequency $\omega_a$ and $\omega_b$) interacts with atoms. The frequency detuning between the ground states $|1\rangle$ and $|2\rangle$, and the excited states $|3\rangle$ is defined as $\delta_{13}(= \omega_a - \omega_{13})$ and $\delta_{23}(= \omega_b - \omega_{23})$, respectively, where $\omega_{13}$ and $\omega_{23}$ are the frequency splitting between the ground states $|1\rangle$ and $|2\rangle$, and the excited states $|3\rangle$, respectively, and the Raman detuning $\delta$ is defined

as $\delta = \delta_{23} - \delta_{13}$. The Rabi frequency $\Omega_{ij}$ for the transition with electric dipole moment $\mu_{ij}$ ($i = 1, 2 \text{ and } j = 3$) is defined as $\Omega_{ij} = \varepsilon\mu_{ij}/\hbar$, where $\varepsilon$ is the electric field amplitude. $\Gamma_3$ is the spontaneous decay rate of the excited state $|3\rangle$, and $\gamma$ is the relaxation rate of the coherence between the hyperfine ground states.

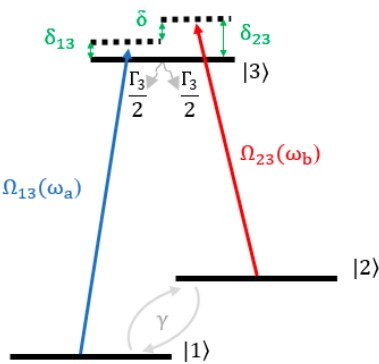

**Figure 1.** Scheme of three-level $\Lambda$ system for CPT interrogating.

Nowadays, technology that uses magneto-optical trap (MOT) to cool and trap neutral atoms is popular. To reach a lower temperature beyond the single-photon Doppler limit, two-photon transition (TPT) was also used for further cooling processes [17–22]. The optical frequency combs (OFCs) correspond to a train of ultrafast pulses generated by mode-locked (ML) laser in the time domain. This laser connects optical frequency with microwave frequency and could provide the purest microwave frequency source [23]. Due to its narrow linewidth and coherent frequencies, OFCs not only are suitable for frequency standards based on CPT [24–26] but also can be used to cool atoms through the TPT process [22]. There is a potential to combine these two functions of the combs together in the cold atomic clock. The repetition rate of OFCs could be locked to the cooled $^{87}$Rb TPT resonance too and has served as a frequency standard, with the stability reaching $1.5 \times 10^{-13}$ at an averaging time of 100 s [27]. This paper proposes a further cooling scheme using OFCs. This cooling scheme utilizes the direct two-photon transitions (DTPT) cooling method during the stimulated emission process to obtain a lower cooling temperature. In the present paper, we study the applications of this kind of cooling scheme on the cold atomic clocks to obtain better frequency stability in the future. In previous studies about atom cooling with OFCs, the cooling processes mainly involved single-photon interactions with the atoms [28–37], which used only a small fraction of the laser's total power and output spectrum. On the contrary, the DTPT cooling method for the atom cooling involves all of the comb teeth contributing together [38].

## 2. Two-Photon Cooling Model by Pulses

We introduced a stimulated emission scheme for the two-photon cooling method by pulses, as shown in Figure 2. A train of pulses generated by ML laser propagating toward the right side interact with the atoms moving to the left side. In momentum–time space, the atoms initially locate at ($P_0$, $t_0$) with momentum $p_0$ toward the left side, and after they absorb TPs with wave vectors $k_1$ and $k_2$ coming from the left side, the atoms go to the final state $|f\rangle$ from the initial state $|i\rangle$ directly through the process of DTPT while losing the momentum of $\hbar (k_1 + k_2)$. Since the excited atoms do not absorb resonantly, their return to the ground state without the spontaneous emission needed for further absorption can only occur by stimulated emission [39,40]. If the stimulated emission is caused by a counter-propagating pulse toward the left side, the excited atoms at ($P_1$, $t_1$) would lose another momentum of $\hbar (k_3 + k_4)$; hence, the total momentum lost is $\hbar (k_1 + k_2 + k_3 + k_4)$. These stimulated processes repeat at the Rabi frequency $\Omega_{if}$ that is much larger than the decay rate $\Gamma_f$. Because the excited atoms return to their ground states more frequently during the stimulated emission process rather than during the spontaneous emission process,

the pressure radiation force from photons will not be in a saturated state at a momentum exchange rate limited by $\Gamma_f$.

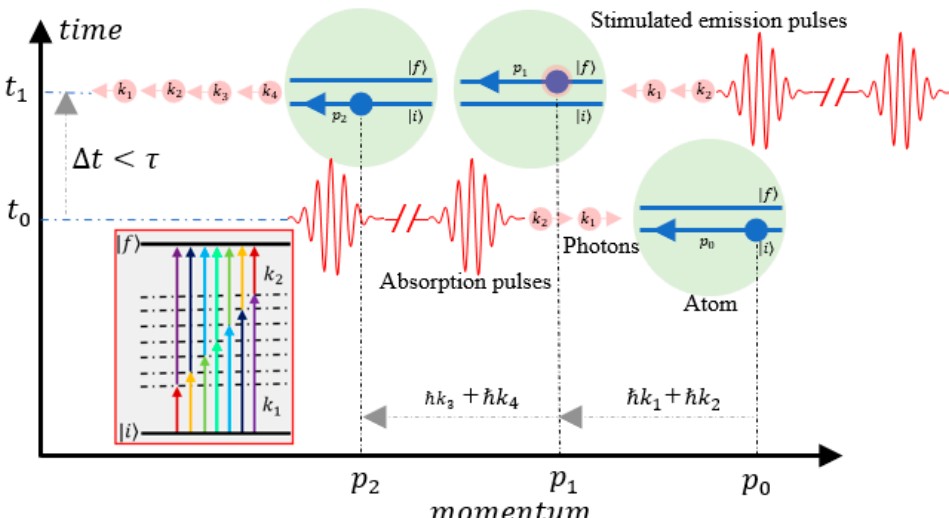

**Figure 2.** Stimulated emission scheme in the momentum–time space, the atoms at $(p_0, t_0)$ lose a momentum of $\hbar\,(k_1 + k_2)$ during the TP absorption process (inset); after a time delay of less than the lifetime $\tau$, the excited atoms at $(P_1, t_1)$ lose another momentum of $\hbar\,(k_3 + k_4)$ during the TP stimulated emission process, and the total momentum transfer or loss of the atoms is $\hbar\,(k_3 + k_4 + k_1 + k_2)$ in one absorption-stimulated emission cycle.

By using the scheme of "absorption-stimulated emission", the recoil momentum obtained by atoms or molecules mainly comes from the fast "absorption-stimulated emission" cycle. By choosing appropriate laser parameters, the rate of this transition cycle can be made much larger than the spontaneous radiation rate, so that the force on the atom or the molecule obviously exceeds the spontaneous radiation force, thus ensuring a shorter deceleration distance and a larger deceleration efficiency. In addition, the experimental setup required by this scheme is relatively simpler. The advantage of this scheme of stimulated radiation deceleration has been confirmed in previous experiments [41] and was used to cool [42] or accelerate [39] atoms/molecules, depending on the details of experiment designs.

Assuming that the atoms are initially in the ground state, the density operator is $\hat{\rho}_a(0) = |i\rangle\langle i|$, and $n$ is used to represent the number of photons in the laser field. In the form of dressed states, the atom-photon density operator is written as

$$\hat{\rho}(0) = \begin{pmatrix} |i,n\rangle\langle n,i| & 0 \\ 0 & 0 \end{pmatrix} \tag{1}$$

We define the pulse area as

$$\theta = \int_{-\infty}^{+\infty} \Omega_{if} g(t)\,dt \tag{2}$$

where $g(t)$ is the pulse envelope and $\Omega_{if}$ is the two-photon Rabi frequency.

The absorption pulse operator can be written as

$$\hat{U}_1 = \begin{pmatrix} \cos\frac{\theta}{2}|i,n\rangle\langle n,i| & -i\sin\frac{\theta}{2}|i,n\rangle\langle n-2,f| \\ -i\sin\frac{\theta}{2}|f,n-2\rangle\langle n,i| & \cos\frac{\theta}{2}|f,n-2\rangle\langle n-2,f| \end{pmatrix} \tag{3}$$

After the pulse is applied, the density operator of the atoms becomes

$$\hat{\rho}_1 = U_1\hat{\rho}(0)\hat{U}_1^\dagger \tag{4}$$

Considering the fact that the atoms in the excited state undergo spontaneous emission at the same time, the atoms in the excited state exponentially decay between the two trains of pulses by a factor of $e^{-t/\tau}$ before the stimulated emission pulse is applied.

The stimulated emission pulse operator is written as

$$\widehat{U}_2 = \begin{pmatrix} \cos\frac{\theta}{2}|i, n+2\rangle\langle n+2, i| & -i\sin\frac{\theta}{2}|i, n+2\rangle\langle n, f| \\ -i\sin\frac{\theta}{2}|f, n\rangle\langle n+2, i| & \cos\frac{\theta}{2}|f, n\rangle\langle n, f| \end{pmatrix} \tag{5}$$

The density operator of the atoms after the stimulated emission pulse becomes

$$\hat{\rho}_2 = \widehat{U}_2\hat{\rho}_1\widehat{U}_2^\dagger \tag{6}$$

By tracing the final density operator, the probability of the population distribution in the ground state ($P_g$) and the excited state ($P_e$) after the two trains of pulses can be obtained as shown in Equation (8) and Figure 3.

$$P_g = \mathrm{tr}_n(\langle i|\hat{\rho}_2|i\rangle) = \cos^2\frac{\theta}{2} - e^{-\frac{t}{\tau}}\sin^2\frac{\theta}{2}\cos\theta \tag{7}$$

$$P_e = \mathrm{tr}_n(\langle f|\hat{\rho}_2|f\rangle) = \left(1 + e^{-\frac{t}{\tau}}\right)\sin^2\frac{\theta}{2}\cos^2\frac{\theta}{2} + \left(1 - e^{-\frac{t}{\tau}}\right)\sin^4\frac{\theta}{2} \tag{8}$$

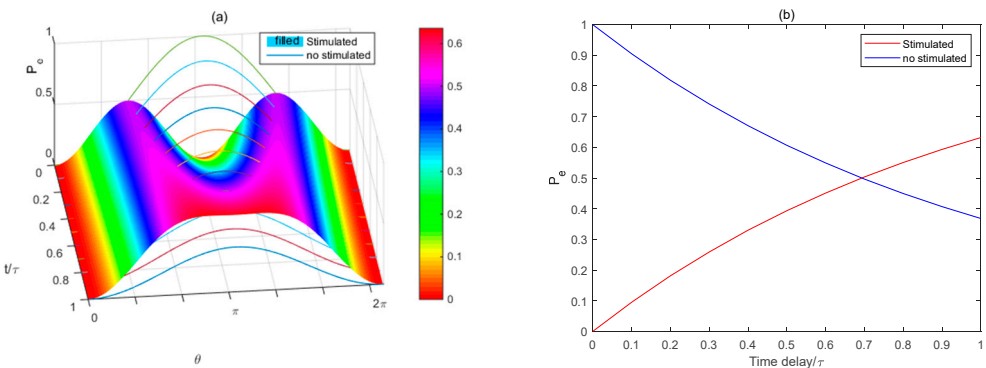

**Figure 3.** The probability of the population distribution in the excited state varies with the pulse area and the time delay of the stimulated emission pulse simultaneously. The curved surface graph in (**a**) and the red line in (**b**) indicate the case in which the stimulated emission pulse is applied, whereas the curved line graph in (**a**) and the blue line in (**b**) indicate that there is no stimulated emission pulse.

The repetition of the period of pulses is the time delay, which is usually less than the lifetime $\tau$ of the atomic excited state. As apparent in Figure 3, the population distribution of the excited state is simultaneously controlled by both the pulse area $\theta$ and the time delay $t$ of stimulated emission pulse. When the pulse area $\theta = \pi$, the highest probability is obtained as shown in Figure 3a. Additionally, in order to keep the high probability, the stimulated emission pulse should be applied at a time delay between $0.7\tau$ and $\tau$, as shown in Figure 3b. If the stimulated emission pulse is applied too early, the excitation probability will be low.

## 3. Results

To show the cooling effect of the stimulated emission pulse on the atoms more directly and by employing the concept of entropy, we compared the temperature change before applying the proposed cooling scheme with that after applying the proposed cooling scheme.

The relationship between entropy operator $\hat{\eta}$ and density operator $\hat{\rho}$ is

$$\hat{\eta} = -\ln\hat{\rho} \tag{9}$$

The ensemble of the entropy operator $\hat{\eta}$ is multiplied by Boltzmann's constant $k_B$ to obtain the corresponding macroscopic quantity information entropy $S$:

$$S = k_B\langle\hat{\eta}\rangle = -k_B\mathrm{tr}(\hat{\rho}\ln\hat{\rho}) = -k_B\sum_n(\hat{\rho}\ln\hat{\rho})_{nn} \tag{10}$$

For our atomic systems,

$$S = -k_B\left(P_g\ln P_g + P_e\ln P_e\right) \tag{11}$$

When $N$ particles are confined to a volume $V$ at a temperature $T$, we have [43,44]

$$S = Nk_B[\ln\left(\frac{V}{N}\right) + \frac{3}{2}\ln T + \sigma_0] \tag{12}$$

Which is based on assuming that $T_1$ and $S_1$, and $T_2$ and $S_2$ are the temperature and the entropy of the atomic group before and after the stimulated emission pulse is applied. According to Equations (11) and (12), the ratio ($R$) of the temperature obtained after applying the proposed cooling scheme to the ratio obtained before applying the proposed cooling scheme can be obtained as

$$R = \frac{T_2}{T_1} = \frac{2}{3}e^{S_2-S_1} \tag{13}$$

Figure 4a shows that the temperature ratio varies with the time delay, and each line has a corresponding pulse area. When the pulse area is small ($\pi/10$), the time delay has little effect on the result, and when the pulse area is $\pi/2$, the ratio increases with the time delay. When the pulse area becomes larger, it can reduce the ratio to the minimum when faced with a proper time delay, such as for $2\pi/3$ pulse area, and the ideal time delay is 0.4. For $\pi$ pulse area, the ratio is independent of the time delay. Figure 4b shows the variation of the temperature ratio with the pulse area, and each line has a corresponding time delay. When pulse area $\theta < \pi/2$, the shorter the time delay, the lower the ratio will be. When pulse area $\theta = \pi$, the ratio does not depend on the time delay, and the ratio is the lowest, which means the lowest temperature could be obtained at pulse area $\theta = \pi$.

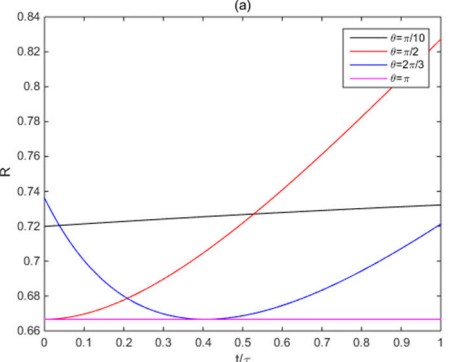 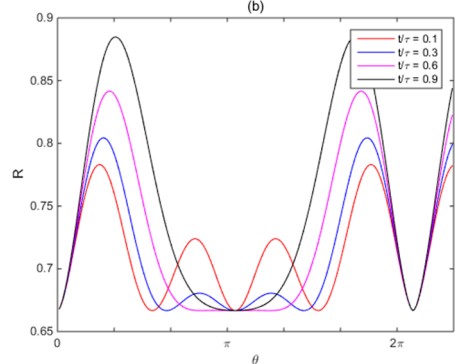

**Figure 4.** The ratio $R$ temperature obtained before and after the application of the stimulated radiation pulse is shown as a function of (**a**) the time delay, wherein the lines each have pulse areas of $\pi/10$, $\pi/2$, $2\pi/3$, and $\pi$ and (**b**) the pulse area, wherein the lines each have time delays of $0.1\,\tau$, $0.3\,\tau$, $0.6\,\tau$, and $0.9\,\tau$.

Considering both the high transition probability and the low temperature obtained, it is best to apply the stimulated emission pulse with the pulse area of about $\pi$, where the time delay is between $0.7\tau$ and $\tau$, such that the temperature of the atom can be decreased to the minimum. As the ML-based DTPT cooling could reduce the temperature to a value much lower than that achieved by the single-photon Doppler [45] and the stability of cold atomic clock is thus improved at the lower temperature, it is predicted that the frequency stability

of cold atom clock based on CPT is will reach a better performance than the previous or current level.

The primary advantages of stimulated pulse together with a DTPT process for atom cooling arises from the use of stimulated emissions in place of spontaneous emissions to return atoms to their ground states. Therefore, the proposed cooling scheme allows for rapid and repeated momentum exchanges between the atom and the light field by restricting the atom–light interaction to a shorter time, compared to the traditional way in which a cycle of absorption is followed by natural decay.

Because of the complex internal structure of some kinds of atoms and molecules, the cw lasers used during single-photon Doppler cooling are not suitable for them, especially in the deep ultraviolet (UV) range, as the single-photon laser cooling is not currently available for the most atoms [45]. For instance, single-photon laser cooling is not suitable for the 1S-2S transition of H atoms for the following two reasons, first, that there is no intermediate real energy level between 1S and 2S of H atoms, and second, that there is no commercially available deep UV laser that will generate 243 nm or shorter wavelength to drive the transition.

However, DTPT cooling with stimulated emission scheme proposed herein can be used to cool these kinds of atoms and molecules. The DTPT process played a main role in interrogating the 1S-2S transition of H atoms [46]. In contrast to cw lasers, OFCs generated by ML lasers can be efficiently frequency-multiplied to apply to the UV and can be used to control the motion of atoms and molecules in ways that are not possible in cw lasers.

## 4. Discussion and Application

When the DTPT stimulated emission cooling method is used in the $^{87}$Rb atomic frequency standard, we can calculate the laser source characteristics required. ML laser electric field strength can be written in the form $\varepsilon(r, t) = \varepsilon_0 g(t) \exp\left(-r^2/\omega_0^2\right)$, where $r$ is the radial distance from the central axis, $\omega_0$ is the beam waist radius, and $\varepsilon_0$ is the peak amplitude of electric field at the central axis. Assuming that the driving pulse has a Gaussian envelope $g(t) = \exp\left[-t^2/2T_p^2\right]$, and $T_p$ is the pulse width, it can be calculated that the time average laser power of the $\pi$ pulse condition is $\overline{p}_\pi = \frac{\sqrt{2\pi^3}\hbar\omega_0^2 f_r \omega_{if}^3}{24\Gamma_f T_p c^2}$. For the Gaussian pulses from the ML lasers, the beam waist radius is $\omega_0 = 20$ μm, the pulse width is $T_p = 1$ ps, the comb tooth repetition frequency is $f_r = 100$ MHz, $\Gamma_f$ is the decay rate of the upper state, and $\omega_{if}$ is the resonance frequency between the upper and lower states. In a popular MOT, a 780 nm laser source is often used, the upper energy level is $5P_{3/2}$, and the laser wavelength that creates the CPT resonance is 795 nm with the upper state of $5P_{1/2}$ due to which the hyperfine energy levels are easily distinguished. When the TPT process is used for cooling, the upper energy level can be $5D_{5/2}$ or $5D_{3/2}$, and a total of 39 pathways and 14 transitions can be identified for the $5S_{1/2}$-$5D_{3/2}$ and $5S_{1/2}$-$5D_{5/2}$ two-photon resonances [22]; the relative transition probabilities are shown in Figure 5. We compared the difference between circularly polarized photons and linearly polarized photons. As shown in Figure 5, the $5S_{1/2}$-$5D_{5/2}$ transition from F = 2 to F = 4 excited by circularly polarized photons has the largest transition probability and can be used for DTPT stimulated emission cooling.

We take the line width of the upper state 5D as 660 kHz and the equivalent DTPT resonance frequency as $7.7 \times 10^{14}$ Hz ($5S_{1/2}$-$5D_{5/2}$). The time average power of the Gaussian $\pi$ pulse generated by ML laser is approximately 420 mW and the peak amplitude of the pulse is about $5 \times 10^5$ V/m, such that the temperature obtained from the DTPT cooling scheme is supposed to reach one tenth of that of the single-photon Doppler cooling limit, which is 145 μK for $^{87}$Rb [45].

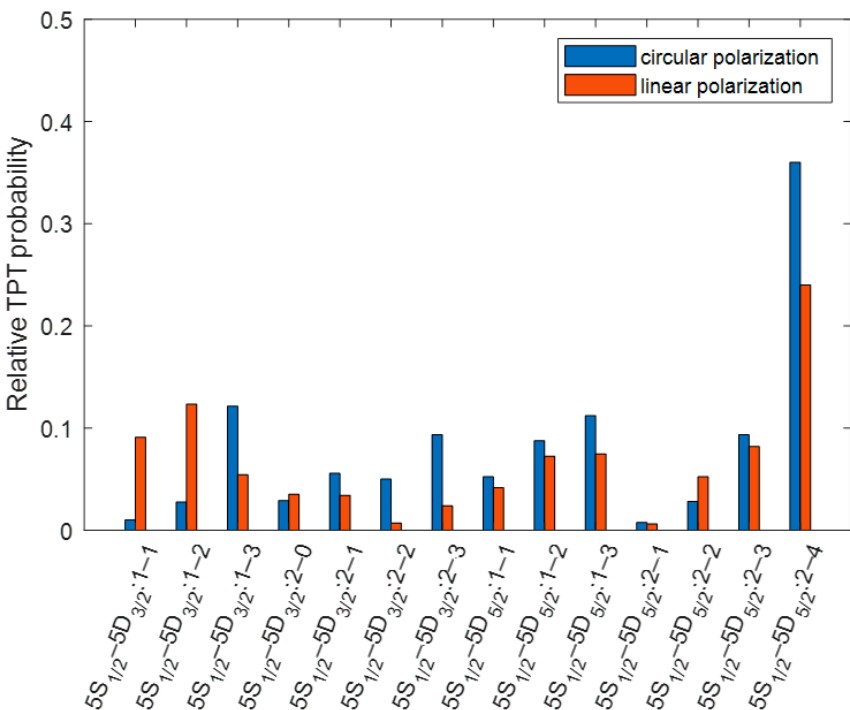

**Figure 5.** Relative probabilities of $^{87}$Rb 5S-5D two-photon transitions, where blue represents excitation with circularly polarized photons and red represents excitation with linearly polarized photons.

Figure 6 illustrates the results of these pulses interacting with the same sample of cold $^{87}$Rb atoms to interrogate the CPT signals based on the configuration in Figure 1. For both cases, the Doppler shifts $\delta_D$ are assumed to be $0.01\Gamma_3$. In the calculation, we used 500 Gaussian pulses to simulate the optical combs, the pulse–atom interaction is described by the Bloch equations as in Ref. [47], and the temporal evolution of the elements $\rho_{ij}$ of the atomic density matrix can be numerically resolved by using the fourth-order Runge–Kutta method. The Rabi frequency is defined as $\Omega_{ij}(t) = \varepsilon(t)\mu_{ij}/\hbar$, for ML, $\varepsilon(t) = \varepsilon_0 \sum_n g(t - nT_r)$, $\varepsilon_0 = 5 \times 10^5$ V/m, the pulse width is 1 ps, and the pulse repetition period is 10 ns. For cw, the Rabi frequency is $\Omega_{13} = \Omega_{23} = \Gamma_3$; $\gamma = 10^{-6}\Gamma_3$; and for the $^{87}$Rb atom $\Gamma_3 = 37$ MHz. We assumed $\rho_{33} + \rho_{22} + \rho_{11} = 1$ and $\rho_{22} = \rho_{11} = 0.5$ under the initial condition to obtain the results shown in Figure 6a. The red, green, and black lines in the figure represent the population of excited state, bright state, and dark state changing with the number of pulses, respectively, where the bright state and dark state are defined as $\rho_B = (\rho_{11} + \rho_{22})/2 + \mathrm{Re}(\rho_{12})$, $\rho_D = (\rho_{11} + \rho_{22})/2 - \mathrm{Im}(\rho_{12})$, respectively. From Figure 6a, we can see that the bright and dark states are at 50% each in the beginning. When the atoms interact with about 40–50 pulses, the bright state (green line) is slowly pumped to the dark state (black line), while the atomic excited state (red line) population slowly becomes 0, the atoms have minimum fluorescence, which is in the dark state, and the CPT phenomenon occurs. We denote the difference from the maximum value of $\rho_{33}$ to the value at the end of 50th pulse as $\Delta\rho_{33}$. The change in the population of excited states during this process corresponds to the process of the fluorescence changing from bright to dark.

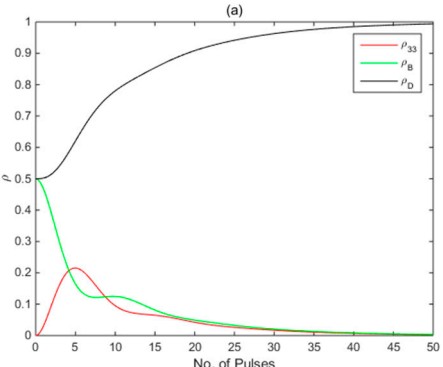 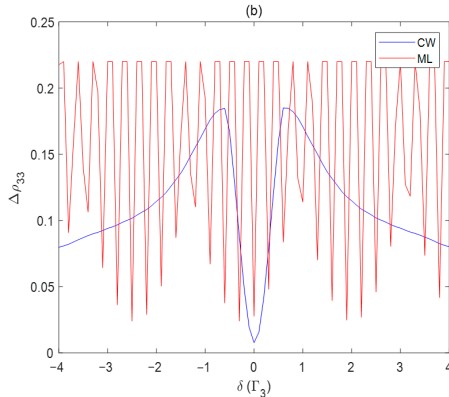

**Figure 6.** (**a**) The population of excited (red), bright (green) and dark (black) states change with the number of pulses. (**b**)The excited state population change ($\Delta\rho_{33}$) of the cold $^{87}$Rb atoms varies with the Raman detuning ($\delta$), and the Doppler shift is set to be $\delta_D = 0.01\Gamma_3$. The interrogating laser fields are cw (blue) and ML (red).

By scanning the Raman detuning $\delta$, we obtained the variation of $\Delta\rho_{33}$ changing with $\delta$ as shown in Figure 6b. The blue line is obtained by using cw, and the red line is obtained using the pulses discussed above. We obtained the shape and linewidth/contrast value of the pulse-excited CPT signal by numerical simulation and compared it with that from the cw-excited signal. From Figure 6b, we calculated the linewidth/contrast values of 0.69 and 0.11 from cw and ML excitations, respectively, as shown in Table 1. As the frequency stability based on CPT is proportional to the linewidth/contrast value, it can be seen that the frequency stability based on CPT in the cold $^{87}$Rb atoms has the potential to be improved by more than six times when using proposed pulses instead of cw.

**Table 1.** The linewidth/contrast values of CPT signals based on different schemes and Doppler shifts.

| Schemes | $\delta_D$ | Linewidth/Contrast |
|---|---|---|
| cw CPT | $\Gamma_3$ | 1.06 |
| cw CPT | $0.1\Gamma_3$ | 0.70 |
| DTPT cooling + cw CPT | $0.01\Gamma_3$ | 0.69 |
| DTPT cooling + ML CPT | $0.01\Gamma_3$ | 0.11 |

As discussed above, the OFCs from ML lasers can be used to cool and interrogate atoms in the one cold atom clock system. Traditional cold atomic clocks usually require three semiconductor lasers for atom cooling, repumping, and CPT interrogation, and each laser requires additional equipment for frequency locking. The entire optical system is complex and bulky. As current mode-locked lasers can easily cover the wavelengths required for Rb atomic cooling, repumping, and CPT detection and can also provide watt-level power output, a single mode-locked laser can be used for atomic cooling, repumping, and CPT detection, which is beneficial to the integration of the optical system. The ML laser parameters depend on the specified element and its corresponding transition energy levels that we study. $^{87}$Rb, for example, during the cooling process, the upper energy level can be $5D_{5/2}$, and the lifetime in this situation is about $\tau = 240$ ns. According to the conclusions drawn in this paper, the time delay can be selected from 170 ns to 240 ns, and the specific delay measures can refer to the method in ref. [39]. $^{85}$Rb can also be used as the cooling element studied, as selected in the literature [45]. If $^{133}$Cs is selected, the upper energy level can be $6D_{3/2}$, $6D_{5/2}$, or $8S_{1/2}$, and the corresponding lifetime $\tau$ is different. As the optical comb frequency is determined by the initial frequency and repetition frequency, the repetition frequency is set as 100 MHz in our calculation and the specific transition can also meet the requirements by selecting the appropriate initial frequency parameters.

## 5. Conclusions

In conclusion, as the ratio value of linewidth/contrast of CPT signal was reduced as the temperature of atoms decreased; hence, the performance of cold atom clock improved. We proposed a more efficient cooling scheme, which utilizes the DTPT process of the OFCs and stimulated emission pulse, to cool atoms for the cold atomic clock to a lower temperature. It was found that the temperature of the atomic sample could be reduced to the minimum when the pulse area was about $\pi$ and the time delay was $0.7\tau{\sim}\tau$, which was dependent on the specified element and its corresponding transition energy levels that we studied. We also calculated the pulse power required for the corresponding $^{87}$Rb cooling process. Additionally, this cooling scheme could be used to cool other elements that cannot be cooled to a desired temperature by traditional single-photon cooling methods. When compared with traditional single-photon cooling methods using cw, if the same optical combs were used to interrogate the CPT signal and when the proposed cooling scheme was utilized, the ratio value of the linewidth/contrast decreased by more than six times, and so did its frequency stability. The optical system of this OFC-based cold atomic clock was integrated and will have potential application in the future.

**Author Contributions:** Conceptualization, L.D.; Data curation, P.G.; Formal analysis, L.D., H.X. and P.G.; Methodology, L.D. and H.X.; Project administration, J.Z.; Resources, J.Z.; Software, H.X.; Supervision, J.Z.; Writing—original draft, L.D.; Writing—review and editing, L.D. and P.G. All authors have read and agreed to the published version of the manuscript.

**Funding:** This research was funded by the National Science Foundation of China, grant numbers 91836301 and 61535001.

**Institutional Review Board Statement:** Not applicable.

**Informed Consent Statement:** Not applicable.

**Data Availability Statement:** The data that support the findings of this study are available upon reasonable request from the authors.

**Conflicts of Interest:** The authors declare no conflict of interest.

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
