# Peer review of "Optical Frequency Comb-Based Direct Two-Photon Cooling for Cold Atom Clock"

_photonics, doi:10.3390/photonics9040268_

Round 1
Reviewer 1 Report
The manuscript “Optical Frequency Combs-based Direct Two-photon Cooling for Cold Atom Clock” by L. Dan us devoted to theoretical (numerical) analysis of the common A-type three level system interaction with optical frequency combs.
Generally I think that the manuscript contains enough of novelty and may consist an interest for a reader.
Unfortunately the organization of the manuscript is a little bit chaotic:
- there is no sense in figure 1b and equations (1) in the introduction.
- Why the second part named “materials”?
- Why figures 3 and 4 are “method”, figure 5 is “result” and figure “6” is discussion?
Besides, personally I think tick labels in figures 1b,3b,4ab,6ab are tiny, while the figures themselves are huge for a such simple curves.
Minor corrections:
- Legend in figure 3a is not obvious, I would recommend to write above blue-bar “filled”.
- Equations (9)-(12) function “ln” should not be in italic.
- Actually the equation4-6 also contain “sin” and “cos” function writte in italic, that makes difficult to distinguish a function and an argument.
I believe that the manuscript deserve to be published in “photonics”, but it should be re-organized.
Reviewer 2 Report
The submitted manuscript studies ways of stability improvements of atomic clocks relying on coherent population trapping. I can share the following observations on this subject:
- The manuscript proposes a new atom cooling configuration that uses counter-propagating pulses with a certain delay between them. It is maintained that implementation of such a configuration will bring a six-fold improvement in atomic clock stability. This result comes from modelling alone and has no experimental verification. Consequently, the reported results raise doubts. The performed numerical calculations have to be corroborated by experiment.
- It is necessary to specify the required pulse duration, is it the nanosecond range, longer, or shorter? The duration of these pulses is an indication of how difficult it will be to produce them (i.e. it will clarify the required performance of the radiation source).
- The Conclusion notes that the entire optical system may be simplified, since a new laser is not needed. But this claim that the optical system may be simpler does not agree with the proposed cooling configuration, in which extra-stringent requirements are imposed on the laser radiation: it must be pulsed (and the shorter the pulses are, the more difficult it is to generate them), a specific delay must be created between the pulses (which will be more or less difficult depending on the pulse duration and the amount of delay). The Conclusion should be corrected.
This manuscript may be published in Photonics on condition that the Authors address the above-listed issues in a further revision
Round 2
Reviewer 2 Report
I have received the Authors’ answers to the presented concerns in their messages, but the revised article itself addresses them only in part. The Authors specify the required radiation parameters—1-ps pulse duration and 100-MHz repetition rate—but laser diodes with such output parameters are unknown (they are either very exotic or totally absent from the market). Consequently, the proposed configuration needs radiation, which is not available from diode lasers. This fact should be commented upon in the manuscript. At the end of Section 4, it is stated that “no additional diode laser is needed for the CPT interrogation, and so it could have more widely application in the future”. It is not immediately clear what Authors mean, why “no additional diode laser is needed”. It is precisely the other way around, because instead of a conventional CW VECSEL, a special radiation source is necessary that would provide 1-ps pulses at 100 MHz. This statement at the end of Section 4 should be reconsidered.
Author Response
Thank you for your advances. We didn't describe it clearly in the previous manuscript. The pulses with the given parameters are generated by a mode-locked laser, not a semiconductor laser. And a ML laser is easy to generate a train of pulses with width of 1 ps and a repetition frequency of 100 MHz (refer to the references: Ann. Phys. (Berlin) 525, No. 7, L29–L34 (2013); Opt. Express 17, 9183-9190 (2009)). There are many such kind of ML lasers that are commercially available, and the pulse width and spectral range can also be adjustable. We rewrite the previous conclusions at the end of Section 4 that are prone to misunderstandings as follows:
Traditional cold atomic clocks usually require three semiconductor lasers for atom cooling, repumping and CPT interrogation, and each laser requires additional equipment for frequency locking. The entire optical system is complex and bulky. As current mode-locked lasers can easily cover the wavelengths required for Rb atomic cooling, repumping and CPT detection, and also can provide watt-level power output, a single mode-locked laser can be used for atomic cooling, repumping and CPT detection, which is beneficial to the integration of the optical system. It would have potential application in the future.